

# Assessing the carbon footprint of the 15th International Coral Reef Symposium (ICRS) in Bremen, Germany

Sophie Littke, Jennifer Bogun and Christian Wild

Marine Ecology Department, Faculty of Biology and Chemistry, Universität Bremen, Bremen, Germany

## ABSTRACT

International scientific conferences serve as vital platforms for networking, knowledge exchange, and collaborative responses to global challenges. To exemplify sustainable practices, it is essential to organize these events in a climate-friendly manner, particularly for topics directly tied to environmental conservation, such as coral reef research. The 15th International Coral Reef Symposium (ICRS), held in Bremen, Germany, in July 2022, aimed to minimize its environmental impact through a comprehensive green strategy. This included reducing $CO_2$ emissions and offsetting unavoidable emissions *via* investments in climate protection projects. This study presents a detailed assessment of the symposium's carbon footprint over its five-day duration, accounting for both travel-related and local emissions. Total $CO_2$ emissions were estimated at approximately 1,491 tons, with transportation contributing 84% and local activities 16%. Local emissions were reduced through measures such as the use of renewable energy, sustainable catering, and waste reduction. The hybrid format of the conference, which enabled online participation, significantly lowered *per-capita* emissions compared to full in-person attendance. The findings highlight the importance of integrating green strategies and virtual participation options to reduce the environmental impact of scientific gatherings. By offsetting all unavoidable emissions, the 15th ICRS successfully achieved carbon neutrality, setting a benchmark for future conferences to prioritize sustainability and environmental responsibility.

## INTRODUCTION

Scientific conferences serve as an important platform for scientists to meet policy makers, establish collaborations, extend networks, meet potential future employers, and share ideas along with expertise to collectively solve problems. Conferences are therefore of great importance to the scientific community. However, as these are sizeable gatherings of people from all over the world, there are undoubtedly environmental factors that should be considered when organizing such events. It is crucial for scientists to set an example during a climate crisis by reconsidering their scientific approaches and how they affect the environment and prevailing climate. We cannot convey the urgent need for change to society until we acknowledge the significance of minimizing greenhouse gas (GHG) emissions and their effects on ecosystems and climate. Rising levels of GHG

Corresponding author
Sophie Littke,
sophie.littke1@gmail.com

emissions in the atmosphere accelerate anthropogenic climate change as a global stressor that severely impacts ecosystems worldwide (*Raupach et al., 2007*). In particular, coral reefs suffer from degradation due to climate change-induced ocean warming and acidification (*Hoegh-Guldberg et al., 2017*; *Hughes et al., 2017*) and act as early warning systems for climate change consequences. Thus, scientific gatherings, and coral reef-related events particularly, need to consider and assess their carbon footprints.

Portions of this text were previously published as part of a technical report (*Littke, Bogun & Wild, 2022*). The International Coral Reef Symposium (ICRS) is the leading conference on coral reef science, conservation, and management, bringing together scientists, researchers, policy makers, and many more from around the world. Its history goes back more than 50 years. The 15th ICRS took place from July 3rd to 8th 2022 in Bremen, Germany, and was the first in-person ICRS ever to happen in Europe. The event was organized in a hybrid format and hosted a total of 1,052 in-person plus 198 online participants.

After the 14th ICRS virtual event was successfully implemented as a carbon neutral event, the set target for the 15th ICRS was to be organized as sustainable and climate friendly as possible. To achieve this, green strategies consisting of two steps were applied: (1) To avoid and minimize local $CO_2$ emissions and waste wherever possible, and (2) To offset all unavoidable $CO_2$ emissions, particularly caused by the travel of participants, through investment in climate protection projects as a compensation measure.

To achieve these goals, a detailed estimation of all emission sources was required. While travel-related emissions have been analysed in prior studies (*Coroama, Hilty & Birtel, 2012*; *Van Ewijk & Hoekman, 2021*; *Milford et al., 2021*), holistic approaches that consider both travel and local emissions are less common. This study aimed to provide a comprehensive assessment by addressing the following research questions: (1) What were the total emissions of the conference? (2) What was the amount of transportation-related emissions and the emission reduction potential in that category? (3) How high were local emissions and how effective were the green strategies in reducing them? (4) How did emissions from online and in-person participation compare in a hybrid format? (5) How feasible and affordable are high-quality offset programs to balance total emissions?

Based on these questions, we hypothesized that transportation—particularly air travel—would contribute a major share of the total carbon footprint. We also anticipated that the implementation of green strategies and the availability of online participation would substantially reduce both local and *per-capita* emissions compared to conventional in-person-only conference formats. Lastly, we expected that all unavoidable emissions could be fully offset through investment in certified carbon offset programs, thereby demonstrating that achieving carbon neutrality is both feasible and financially attainable for large-scale scientific events.

## MATERIALS & METHODS

To assess the extent of $CO_2$ emissions for the 15th ICRS, a calculation following the greenhouse gas (GHG) protocol (*Russell, 2019*) was carried out. For the estimation of

**Table 1  Greenhouse gas (GHG) protocol emission scopes.** Overview of the three emission scopes and the respective carbon sources considered relevant for the emission calculation of the 15th International Coral Reef Symposium 2022.

|  | Definition | Relevant categories for ICRS 2022 |
|---|---|---|
| Scope 1 | Direct emissions from sources that are owned or controlled by the facility or company | None |
| Scope 2 | Indirect energy related emission including purchased energy that is not produced within the facility | Electricity<br>Heating<br>Cooling<br>Online attendees |
| Scope 3 | All other indirect emissions | Transportation<br>Public transport<br>Catering<br>Accommodation<br>Waste<br>Additional items |

various forms of emissions, the GHG Protocol distinguishes between three scopes (Table 1). The calculation was carried out for 1,052 in-person attendees, including volunteers, the organization team, and the press, plus 198 online attendees for a conference duration of five days. Attendee numbers were obtained from the registration lists provided by the ICRS 2022 organization team. The initial registration list was obtained approximately two months prior to the event and used for detailed calculation of travel-related emissions. An updated registration list was acquired about one month before the event.

Research was undertaken to identify the $CO_2$ emission factors (Table 2) for emissions generated by all listed scope categories. As some of the categories differ in their $CO_2$ emission factors among countries (*e.g.*, hotel accommodation), the pursued strategy was to specifically identify emission factors that were most applicable to Germany. Key words for the research included "$CO_2$ emissions", "carbon dioxide emissions", and "emission factor", specified for the individual categories. In some cases, the determined factors have been converted into consistent units, *e.g.*, miles into kilometres.

## Transportation

Emissions for travel to and from Bremen were calculated using the mean distance between Bremen and each attendee's country of residence (*Georg, 2022*), excluding the attendees residing in Bremen. The attendee's countries of residence were obtained from the conference registration list. No specific airport within a country was chosen; instead, a central location within the respective country was used for distance calculation. Distances greater than 500 km were assumed to involve airplane travel, whereas shorter distances were assumed to be split evenly between train and car travel. Hence, in the calculation, 780 attendees were presumed to use airplane travel, and 131 attendees were presumed to arrive by train or car. Direct flights in economy class without individual carbon offset purchases were assumed in our base calculation for all participants travelling by airplane as no reliable data was available. To assess the potential impact of deviations from this assumption, we retrospectively conducted a sensitivity analysis in which 5–10% of attendees were assumed

**Table 2 $CO_2$ emission factors.** Emission factors used for calculations in all relevant categories including the respective sources.

| Category | $CO_2$ emission factor | Reference |
|---|---|---|
| *Transportation* | | |
| Airplane | 101.3 g/RPK | *EPA (2022)* |
| Car | 130.3 g/km | *EEA (2022)* |
| Train | 64 g/km | *Zheng (2022)* |
| Public transport | 199 g/km | *EPA (2022)* |
| *Food* | | |
| Rice/potatoes | 2,455 g/kg | *Ritchie, Rosado & Roser (2022)* |
| Vegetables | 980 g/kg | *Ritchie, Rosado & Roser (2022)* |
| Cheese/eggs | 14,274 g/kg | *Ritchie, Rosado & Roser (2022)* |
| Milk | 3,150 g/kg | *Ritchie, Rosado & Roser (2022)* |
| Coffee/tea | 28,530 g/kg | *Ritchie, Rosado & Roser (2022)* |
| Wine | 1,790 g/kg | *Ritchie, Rosado & Roser (2022)* |
| Beer | 500 g/pint | *Berners-Lee (2010)* |
| 1.5 l glass water bottles | 323 g/bottle | *Tappwater (2022)* |
| *Accommodation* | | |
| Night/three-star hotel | 16,900 g | *Ratjen (2016)* |
| *Energy* | | |
| Norwegian hydropower | 3.33 g/kWh | *Silva & Saur Modahl (2015)* |
| Tapwater (sanitary use) | 468 g/kWh | *Meunier (2020)* |
| Wastewater | 290 g/m3 | *Wang et al. (2016)* |
| Household waste | 500 g/kg | *BEHG (2020)* |
| Online attendance | 10,000 g/attendee | *Strüber (2021)* |
| *Additional items* | | |
| T-shirts | 8,300 g/t-shirt | *Khan & Islam (2015)* |
| Conference pass | 6 g/pass | *Ezeep (2022)* |
| Lanyards | 2,700 g/kg | *WePrintLanyards (2022)* |

**Notes.**
*RPK, Revenue passenger kilometre.
*EPA, United States Environmental Protection Agency.
*EEA, European Environment Agency.
*BEHG, Brennstoff-Emissionshandelsgesetz (Germany).

to fly business class. Business class travel was estimated to generate approximately three times the emissions of economy class travel (*Bofinger & Strand, 2013*; *Ciers et al., 2018*). Furthermore, no car-sharing was assumed, meaning emissions were calculated as if every attendee arriving by car was driving their own vehicle. This assumption was made to ensure that car-related emissions were not underestimated, thereby fully covering potential emissions. Total emissions caused by travel of participants were established through multiplication with suitable emission factors (Table 2). When the initial transportation emissions were calculated, only 911 in-person attendees were listed on the registration lists. For 141 attendees who registered late, travel-related emissions were assumed based on the average $CO_2$ emissions per previously registered participant.

CO$_2$ calculations for public transport included a two-way journey between the airport and central station for all attendees arriving by airplane. Additionally, it was presumed that 25% of attendees, as well as the organization team, used public transport twice a day between their accommodation and the conference location. This assumption was based on the fact that most attendees from outside Bremen likely resided in hotels within walking distance of the venue, as indicated by the majority of hotels listed in the event organizers' accommodation suggestions. For attendees staying more than one km from the venue or those residing within Bremen, public transport use was assumed. Usage of public transport instead of taxis or other transport modes was presumed as it was encouraged by providing free public transport for the entire conference duration, included in the registration fee.

## Catering

Emission calculations for food and drinks considered a total of six meals, two ice-breaker drinks, and ten coffees with milk per attendee for the entire duration of the conference. Catering was mostly based on vegetarian, regional, and seasonal products. The average size of each meal was assumed to be 500 g, consisting of 200 g rice or potatoes, 200 g vegetables, and 100 g cheese or eggs. Emission factors used for calculation were determined for main ingredient categories individually (Table 2) but were not specific to the catering provider as this information was not obtainable. For shared categories the average emission factor of the respective categories was used. Furthermore, a daily consumption of 1.5 L of drinking water per attendee, provided in 1.5 L glass bottles, was assumed.

## Accommodation

For the calculation of CO$_2$ emissions from hotel accommodation, it was assumed that all in-person attendees from outside Bremen, totalling at 938 attendees, had five overnight stays in hotels with an average star category of three. The three-star category was chosen based on a list of hotels provided by the event organizers to attendees, which suggested accommodations of different categories averaging around three stars. No exact data on accommodation choices was available, but feedback from attendees indicated that the majority of participants from outside Bremen stayed in accommodations recommended on that list, mostly due to the close proximity of these hotels to the venue. It was further assumed that each attendee occupied a single room (no room sharing). The emission factor applied specifically represents overnight stays in three-star category hotel accommodations in Germany.

## Energy, waste and additional items

Energy consumption for all occupied conference halls was calculated for the total span of the conference based on known consumption data provided by the venue operators *via* private email communication. Total energy usage at Congress Halls was estimated at 22,000 kWh based on previous comparable events and 10,163,226 kWh at Congress Centrum Bremen (CCB), based on month-specific data. While some energy was provided by solar panels, exact amounts could not be identified. Therefore, all energy consumption was calculated assuming electricity sourced from Norwegian hydropower to ensure comprehensive coverage of emissions.

**Table 3** **Total $CO_2$ emissions from the 15th International Coral Reef Symposium 2022.** Detailed overview of emissions by category. The energy category comprises all Greenhouse Gas Protocol scope 2 emission sources (electricity, heating, cooling, and online attendees).

| Category | tCO$_2$ |
|---|---|
| Travel of participants | 1,139.8 |
| Catering | 94.2 |
| Hotels | 79.3 |
| Energy | 36 |
| Additional items | 4.4 |
| Public transport | 1.9 |
| Waste | 0.3 |
| Total | 1,355.9 |
| **+ 10% SAFETY MARGIN** | **1,491.5** |

Water consumption for sanitary use, including sewage, was estimated based on three daily lavatory uses per attendee, with 30 s of handwashing each time, plus an additional use during the Icebreaker event. Water consumption per lavatory use, including handwashing, was estimated at 8 L. $CO_2$ emissions related to water provision were calculated based on the arising electricity required (0.51 kWh per m$^3$ of water) (*Baumgarten et al., 2014*).

$CO_2$ emissions generated by online attendees were calculated with the emission factor of 10 kg of $CO_2$ per attendee based on the result of the carbon footprint assessment of the 14th ICRS virtual event (Silke Strüber, Beks EnergieEffizienz, please see here: https://www.icrs2022.de/fileadmin/photos/Green_Strategy/2021-07-22_Calculation_Carbon_Footprint_ICRS_2021.pdf). For waste, an amount of 0.15 kg per meal, added up by one napkin per person per meal and coffee break was estimated. An average emission factor for general household waste in Germany was used for calculation. The calculation for $CO_2$ emissions from additional items was based on numbers provided by the organisation team and encompasses the production of 527 t-shirts and 1,000 conference passes and lanyards.

## RESULTS

### What were the total emissions of the conference?

The calculations showed that the total amount of $CO_2$ emissions produced by the 15th ICRS, including a 10% safety margin, was approximately 1,491 t (Table 3). Of these, most emissions (84.1%) were caused by transportation to and from the conference, whereas all local emission sources combined contributed 15.9% (Fig. 1).

### What was the amount of transportation-related emissions?

With 1,139.8 tCO$_2$, or 83.3%, participant transportation to and from the conference location represented the distinct majority of $CO_2$ emissions. The travel emissions for each of the 1,052 in-person attendees were approximately 1.08 tCO$_2$ per attendee. Within this category, the main emission source was airplane travel to and from Bremen (Fig. 2). Airplane travel accounted for 1,135.7 tCO$_2$, or 99.6%, of all travel-related $CO_2$ emissions assuming only direct flights in economy class. Sensitivity analysis revealed that if 5–10% of

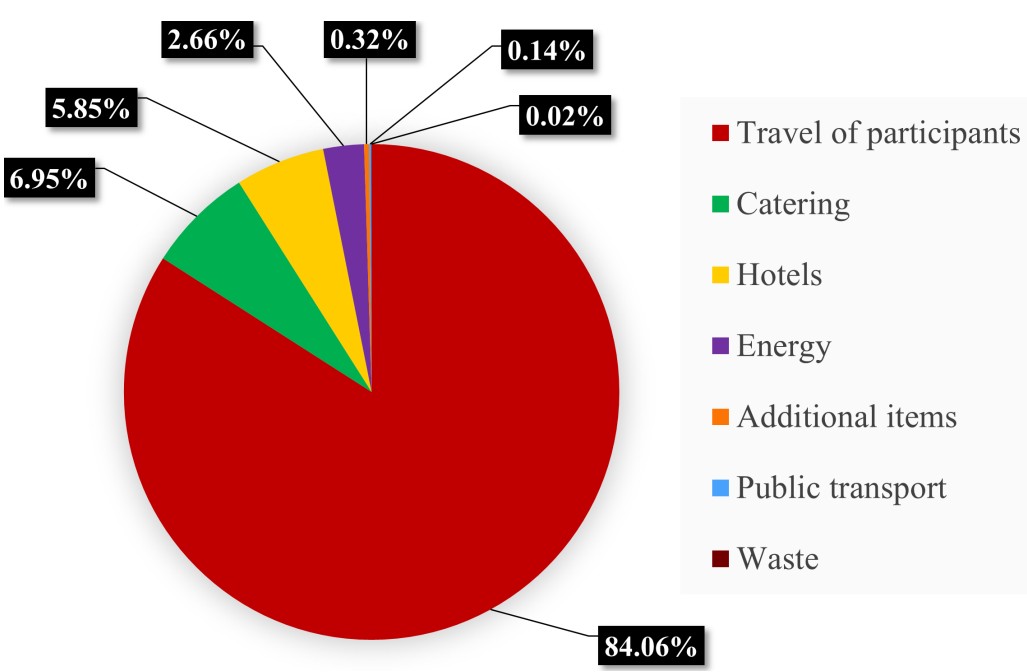

**Figure 1  Total CO$_2$ emissions.** Proportional contribution of key categories to the overall carbon footprint of the 15th International Coral Reef Symposium.

participants flew in business class, flight emissions rose by approximately 104–207 tCO$_2$. Travel by train and car combined contributed only 0.4% to the total emissions.

### How high were local emissions?

Local emission sources accounted for 16.2% of all CO$_2$ emissions. The main emission source within local emissions was found to be catering with 94.2 tCO$_2$, equivalent to 43.6% (Fig. 3), which contributed 6.9 % to total CO$_2$ emissions. This was followed by accommodation of attendees from outside Bremen with 16.9 kgCO$_2$ per person/night for 938 participants. Hotel-related emissions therefore totalled at 79.3 tCO$_2$, equivalent to 36.7% of local emissions. The main category of energy, including electricity, heating, cooling, water usage, sewage, and emissions caused by the online platform, contributed 16.7% to all local emissions. Altogether, emissions arising from additional items (4.43 tCO$_2$), public transport (1.9 tCO$_2$) and generated waste (0.27 tCO$_2$) during the conference, sum up to a total of 3.0% of all local CO$_2$ emissions.

### How do online and onsite-derived emissions compare in a hybrid event?

The total emissions from the in-person event were 1,353.8 tCO$_2$, with *per-capita* emissions of each in-person attendee averaging at 1.29 tCO$_2$. However, emissions from online attendance totalled 1.98 tCO$_2$ for 198 conference participants, contributing only approximately 0.15% to all the emissions of the conference. The *per-capita* emissions for online participants were 0.01 tCO$_2$, which is much lower than the average of 1.29 tCO$_2$ per in-person attendee.

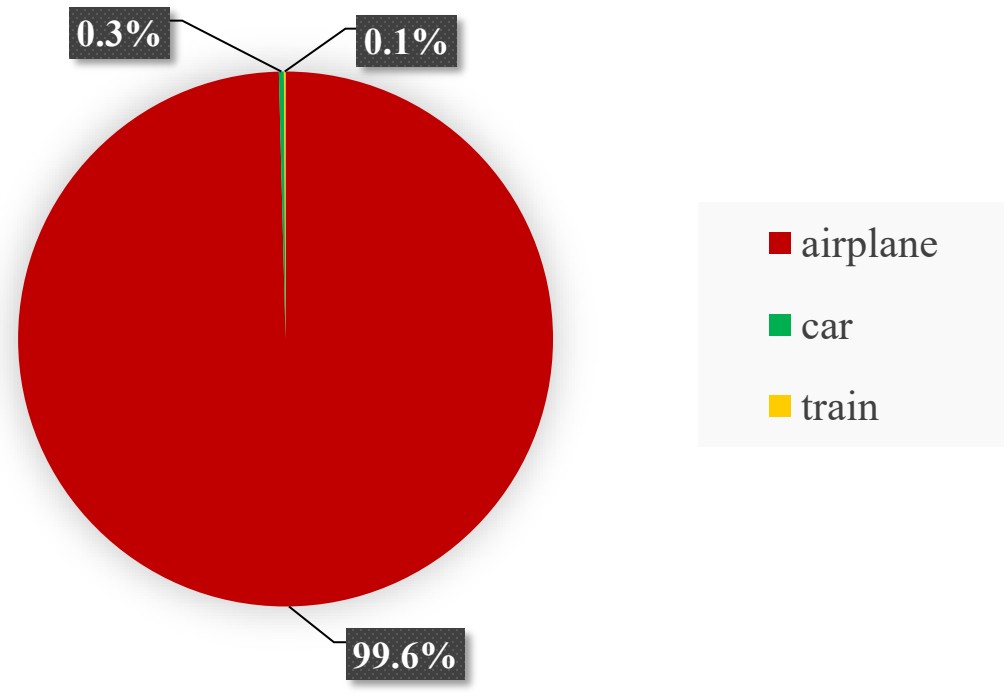

**Figure 2** **Transportation-related emissions.** Proportions of unavoidable $CO_2$ emissions from participants' travel contributed by three subcategories.

### How feasible and affordable are high-quality offset programs to balance total emissions?

To compensate for unavoidable emissions and achieve carbon neutrality, the conference organizers, together with their climate accounting partner KlimaInvest, developed a diverse portfolio of certified remediation projects. These projects were selected based on their geographic distribution, relevance to coral reef ecosystems, alignment with the UN Sustainable Development Goals (SDGs), and adherence to high certification standards. The carbon offset was distributed evenly across five projects, each receiving 20% of the total offset investment. Four of the projects supported renewable energy generation in the Dominican Republic, Mauritius, Aruba, and India, while the fifth focused on protecting mangrove and coastal swamp forests in Borneo, Indonesia (for further details about these projects, please see here: https://www.icrs2022.de/green-strategy#c200). The average cost of offsetting was 15 € per $tCO_2$, which translated to approximately 19.35 € per in-person attendee based on the calculated *per-capita* emissions of 1.29 $tCO_2$.

## DISCUSSION

### What were the total emissions of the conference?

The total carbon footprint of the 15th ICRS was approximately 1,491 $tCO_2$, including a 10% safety margin. This calculation incorporated emissions from multiple sources:

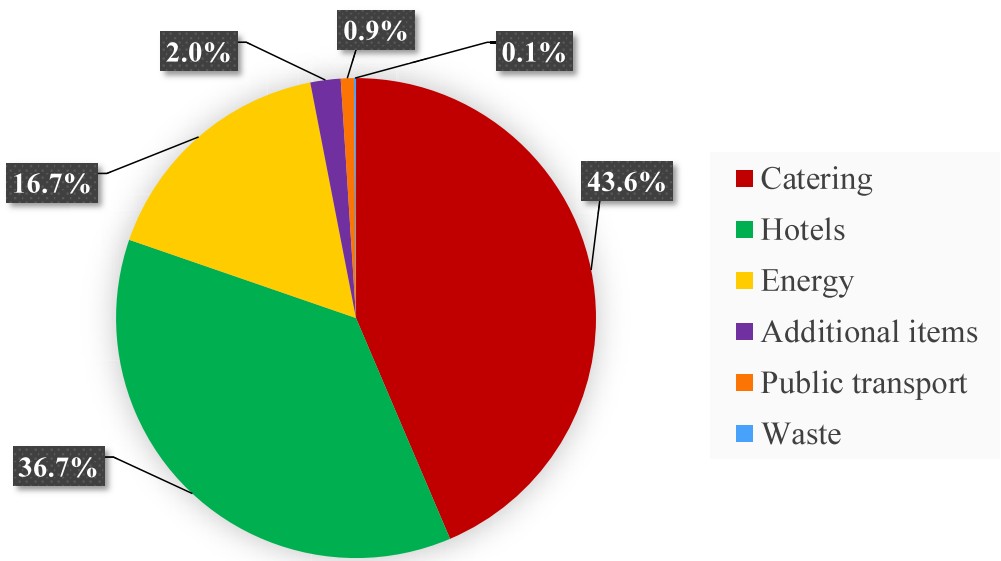

**Figure 3 Local CO$_2$ emissions.** Proportional contributions of the six main categories of local emission sources.

transportation, local activities (*e.g.*, catering, accommodation, venue energy use), and online participation. While most of the emissions were associated with transportation, the analysis highlights the importance of addressing all facets of conference planning to reduce overall environmental impacts.

Comprehensive assessments that account for a wide variety of emission sources remain scarce. Furthermore, variations in conference locations and number of attendees render broad comparisons with other events more challenging. A study that carried out a Life Cycle Assessment of a scientific conference on sustainability held in Germany (*Neugebauer et al., 2020*), estimated their total carbon footprint at 455 tCO$_2$, with *per-capita* emissions of 0.57 tCO$_2$. While this amount is considerably smaller than the results of our calculations, the number of participants was lower (800) than at ICRS. In addition, a vast majority of participants (approximately 85%) originated from Europe, while a larger number of attendees at ICRS travelled from further locations, also explaining higher *per-capita* emissions at ICRS (1.29 tCO$_2$). A hypothetical in-person conference held in Innsbruck, Austria, with 1,500–1,900 attendees, reported a total carbon emission range of 1,249.14–2,575.12 tCO$_2$ (*Jäckle, 2021*). The total emissions of the 15th ICRS fall within the lower end of that range, despite the inclusion of additional categories like waste and online participation. This suggests that emissions at ICRS were relatively moderate for an international event.

The 15th ICRS provides a useful benchmark for understanding the emissions profile of hybrid conferences. Its Central European location, integration of sustainable practices, and hybrid format collectively exemplify strategies for managing emissions while facilitating large-scale scientific engagement.

## How high were transportation-related emissions?

The calculation of the carbon footprint of the 15th ICRS showed that approximately 1,140 $tCO_2$, equalling 83.8% and therefore a vast majority of all emissions, were caused by unavoidable travel of in-person attendees. Travel of participants as the dominant factor contributing to $CO_2$ emissions is in accordance with previous studies that estimated the environmental impact of both, travel and local emissions of academic conferences (*Neugebauer et al., 2020*; *Jäckle, 2021*). The total amount of travel emissions of the ICRS 2022 was higher compared to conferences in Chicago, USA (2017), Surrey, UK (2015) and Ulsan, South Korea (2013) with 955, 765, and 722 $tCO_2$, respectively (*Van Ewijk & Hoekman, 2021*). Travel emissions per in-person attendee at the 15th ICRS were lower (appr. 1.3 $tCO_2$) compared to two of the three previously mentioned conferences (USA: 1.5 $tCO_2$, South Korea: 1.8 $tCO_2$) and equal to the conference held in the UK. This indicates that lower travel emissions per attendee likely resulted from the conference's location in Central Europe, resulting in a relatively high number of attendees arriving by train or car. Consequently, selecting optimal conference locations in the future to reduce travel emissions could help to minimize the environmental impact of scientific conferences.

Retrospective sensitivity analysis regarding the impact of different flight classes showed that emissions from business class travel can be substantially higher than economy class travel due to increased space per passenger and associated services (*Bofinger & Strand, 2013*; *Ciers et al., 2018*). This highlights the importance of including a conservative safety margin and to increase accuracy of future carbon footprint assessments by gathering more detailed information regarding travel class and individually purchased carbon offsets during the registration process.

While the location of the 15th ICRS and the hybrid format supported the decrease in travel emissions, the number of international attendees travelling far routes, coupled with generally high aviation emission factors, still resulted in airplane travel having the highest impact on the carbon footprint by far.

## How high were local emissions?

Compared to transportation emissions, all local emissions combined, contributed a rather small proportion of 16.2% to total emissions. This was realized by following the green strategy of avoiding and minimizing local emissions wherever possible. While emissions originating from international travel were inevitable for an in-person conference bringing together coral reef experts from all over the world, emissions from all other categories were reduced to a minimum. This was achieved through a variety of measures. A conference location that took measures to reduce $CO_2$ emissions arising from energy provision was chosen. This was achieved by using LED lamps in the conference centre and by covering 25% of all energy needs through solar panels, and the rest by Norwegian hydropower stations (MESSE BREMEN).

The second largest contributor to the $CO_2$ emissions of the ICRS 2022 was catering. With a total of 94.2 $tCO_2$, emissions stemming from catering are certainly relevant to the environmental impact of the conference. The number is higher compared to estimated values between 7.09–47.71 $tCO_2$ for a hypothetical event taking place in Innsbruck, Austria

 

(*Jäckle, 2021*). The latter event calculated catering emissions for 1,500–1,900 participants but with a much lower per-person consumption (three lunches, one dinner) than the calculations for the ICRS 2022. Excluding drinks such as coffee and alcoholic beverages could significantly lower the estimation of the hypothetical event in Austria. Overall, a large uncertainty lies in the comparability of the emission factors used for the calculations, as they can vary largely depending on the source of the product. As the conference catering at the ICRS 2022 was focused on sustainable, regional, and mostly vegetarian food, the related $CO_2$ emissions could still be kept relatively low for the large number of participants and the long duration of the conference.

Overnight stays in hotels for participants from outside Bremen contributed 37% of the total $CO_2$ emissions. The emission factor of 16.9 $kgCO_2$ per person/night used for calculation was higher compared to 6.85 $kgCO_2$ per person/night in another study (*Astudillo & Azarijafari, 2018*). These factors can largely vary depending on room types and services of the hotel. In addition, assumptions regarding multiple participants residing in the same room could have largely altered the results. The calculations for the ICRS 2022 assumed a separate room for each participant from outside Bremen to ensure that the calculation result covered the actual $CO_2$ emissions within that category.

Altogether, the categories of additional items, public transport, and waste contributed a rather small proportion of the emissions, with a total of 6%, partly because of efforts to keep local emissions as low as possible. The comparatively small contribution may explain why most studies thus far have not included these components in their calculations. However, to obtain comprehensive estimations, these categories were included in our calculations. The amount of waste produced was reduced by avoiding the use of disposable materials, for example through reusable dishes and cutlery for lunch, dinner, and coffee breaks. To reduce local emissions caused by transportation within Bremen, all registered attendees were given the opportunity to use public transportation free of charge for the entire conference duration. Additionally, the conference took place in the city centre within a short walking distance from the central railway station, as well as many accommodations and restaurants. Furthermore, a t-shirt brand was chosen that designed the entire production and shipment process as eco-friendly and sustainable as possible, for example by using 100% organic cotton.

Overall, local emissions accounted for a smaller, yet still substantial, proportion of the total, with efforts such as sustainable catering and the use of renewable energy at the venue demonstrating the potential for impactful mitigations. This shows that despite their rather small contribution to total emissions, local emission sources remain crucial for overall assessments, as they are often more directly within the control of organizers and can offer immediate opportunities for emission reductions.

### How do online and on-site derived emissions compare in a hybrid event?

As shown from the $CO_2$ calculations of the 14th virtual ICRS, the total emissions per online attendee (0.01 $tCO_2$) were much lower than those caused by in-person attendees, underlining the considerable environmental benefits of virtual participation. These results
align with findings from *Van Ewijk & Hoekman (2021)*, who identified virtual conferences as having the highest potential for reducing travel-related emissions. However, while fully virtual conferences dramatically reduce environmental impacts, they may limit the quality of discussions and exchange of ideas, which are central to scientific collaboration and networking.

Hybrid formats, such as the 15th ICRS, offer a promising compromise. By enabling virtual participation alongside in-person attendance, hybrid conferences significantly lower travel-related $CO_2$ emissions, while preserving opportunities for face-to-face interaction. Future research should focus on optimizing hybrid events to balance inclusivity, professional development, and environmental sustainability. Enhancing virtual engagement and interaction could ensure that online attendees enjoy equitable access to networking and knowledge exchanges, making hybrid conferences an effective model for sustainable scientific collaboration.

### How feasible and affordable are high-quality offset programs to balance total emissions?

By thoroughly collecting data, calculating total greenhouse gas emissions, and purchasing high-quality carbon credits across a diverse portfolio of five offset projects, the ICRS 2022 was able to effectively balance its emissions and reach carbon neutrality. The result of the carbon footprint assessment was verified by climate accounting professionals, and offsets were applied across a balanced portfolio of certified initiatives across a wide global range promoting green energy and the protection of mangroves and coastal swamps. Nature-based solutions (NBS), such as mangrove conservation and restoration, reforestation and soil carbon enhancement, offer a particularly valuable approach, as they not only sequester carbon, but also support biodiversity, ecosystem services, and local livelihoods. For example, mangrove restoration is particularly relevant for the marine science community due to its alignment with coastal resilience and habitat conservation and can therefore enhance the narrative and educational value of sustainability efforts. We recommend that future events continue to support a balanced offset portfolio that includes both nature-based and renewable energy projects, prioritizing high-integrity initiatives that are certified by reputable standards and aligned with the values of the scientific community.

### Future directions for climate responsibility in scientific events

As climate change continues to accelerate, the scientific community has a responsibility not only to generate knowledge but also to model the behavioural shifts needed for mitigation. To that end, we advocate for carbon footprint assessments to become a standard, ideally mandatory, component of future scientific conferences. Mandatory assessments would ensure that all events systematically measure and disclose their climate impact, thereby promoting consistency and encouraging organizers to take active steps toward reduction and compensation. Voluntary approaches, while often well-intentioned, depend heavily on individual initiative and resources and may not always be sufficient.

To support this shift, institutions or scientific societies could offer centralized tools, such as standardized data collection templates, emissions calculators tailored to conference
logistics, and best-practice guidelines, or collaborate with professional services that specialize in event-related carbon accounting. These services could assist in compiling travel data, estimating emissions from catering, accommodation, and venue energy use, and identifying credible carbon offset providers. Future research to improve the accuracy of emission estimates, develop standardized carbon accounting tools and model alternative low-emission formats could further support this transformation. Alongside these measures, ensuring that conference outputs remain accessible would maximize scientific impact and reduce the need for repeat in-person attendance.

Beyond measurement, offset payment schemes raise important questions about responsibility and equity. Embedding a set carbon fee (*e.g.*, to cover 1–2 $tCO_2$ *per-capita* emissions) into the registration cost, can ensure systematic and sufficient offsetting, and shift the burden away from organizers. However, mandatory contributions may unintentionally discourage participation, especially for students or researchers from low-income institutions and countries. Voluntary contributions, while more flexible, can be underutilized and may result in an uneven distribution of responsibility across participants (*Smith et al., 2024*). Future models might consider tiered pricing schemes based on geographic distance or institutional resources, or opt-out models with justification, to balance fairness and effectiveness.

Finally, although carbon-neutral conferences are an important step, they remain a small piece of academia's total carbon footprint. Daily university operations, research expeditions, and commuting are far more persistent emission sources. The success of ICRS 2022 can serve as a framework for conference-level accountability but also highlights the need for broader institutional change. The principles demonstrated at sustainable conferences can inform year-round action by academic institutions and scientific societies. One critical step is implementing year-round carbon accounting frameworks that track emissions from all core activities, including lab operations, travel, and facilities. These should be accompanied by annual climate impact reports to promote transparency and accountability. Sustainable travel policies can also play a central role, encouraging virtual or hybrid attendance for meetings, prioritizing low-emission travel options, and integrating carbon budgets into departmental planning. Beyond operations, institutions could promote low-carbon research practices by incorporating environmental responsibility into grant evaluations, promotion criteria, and recognition awards.

We recognize that not all institutions and regions will have the same capacity to implement these measures. Nonetheless, adapting efforts to local resources and priorities can help the scientific community to move toward a more inclusive and environmentally responsible academic culture. The 15th ICRS marked an important step by achieving carbon neutrality through comprehensive emissions accounting and offsetting. While future ICRS meetings will be organized by different teams, we hope that our methods and results will serve as a foundation for further climate-responsible planning, ideally progressing toward fully net-zero scientific events.

## CONCLUSION

This study presents a comprehensive carbon footprint assessment of the 15th International Coral Reef Symposium (*ICRS, 2022*), highlighting both the challenges and opportunities in reducing emissions from large scientific events. Based on our calculations, all unavoidable emissions were compensated for by investing in climate protection projects. Therefore, the target of implementing the 15th ICRS as a $CO_2$ neutral event was successfully achieved. This achievement was based on the following green strategies: (1) Minimization and avoidance of emissions wherever possible, and (2) compensating all inevitable emissions through investment in climate protection programs.

Overall, this assessment highlights the value of robust carbon accounting as a foundation for climate-conscious decisions in academic event planning. As the scientific community continues to advocate for climate solutions, it should also lead by example, leveraging hybrid formats, sustainable logistics, and verified carbon offsets. Future conferences should prioritize low-emission venues, optimize participation modes, and strive for standardization in emission tracking to support broader climate goals.

## ACKNOWLEDGEMENTS

We would like to thank Andrea Hess and Davina Pick from the Convention Center (CCB) Bremen, Germany, Manou Gernat from MESSE BREMEN, Germany, Peter Sebastian from Maritim Hotel & Congress Centrum Bremen, Germany, Oliver Heitmann from KlimaInvest, Hamburg, Germany, and Silke Strüber from Beks EnergieEffizienz, Bremen, Germany, for their constructive advice and critical review of our calculations.

### Funding

This work was funded by the Baseline funds of the Department Ecology (University of Bremen). The funders had no role in study design, data collection and analysis, decision to publish, or preparation of the manuscript.

### Grant Disclosures

The following grant information was disclosed by the authors:
Department Ecology (University of Bremen).

### Competing Interests

The authors declare there are no competing interests.

### Author Contributions

- Sophie Littke conceived and designed the experiments, performed the experiments, analyzed the data, prepared figures and/or tables, authored or reviewed drafts of the article, and approved the final draft.
- Jennifer Bogun conceived and designed the experiments, performed the experiments, analyzed the data, prepared figures and/or tables, authored or reviewed drafts of the article, and approved the final draft.

- Christian Wild conceived and designed the experiments, authored or reviewed drafts of the article, and approved the final draft.

## Data Availability

The raw data is available in the Supplementary File.

## Supplemental Information

Supplemental information for this article can be found online at http://dx.doi.org/10.7717/peerj.19811#supplemental-information.

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
