# Peer review of "Assessing the carbon footprint of the 15th International Coral Reef Symposium (ICRS) in Bremen, Germany"

_PeerJ, doi:10.7717/peerj.19811_

## Round 0.1 · original submission · Minor Revisions

·

Basic reporting

Clear and relevant hypothesis for assessment. Suggest also adding an item on the implementation of offsets, ie, 5. Feasibility of implementing carbon offset programs to balance total emissions

Experimental design

- Methodology assumes 'Direct flights in economy class were assumed for all participants travelling by airplane'- however, some travellers (assume 5-10%) would be business class, and the GHG contributions are much higher.

- methodology- would some participants have offset flights, e.,g Qantas carbon offset

- The portfolio of remediation projects is discussed in conclusion, but should be included in results with some details on the amount of GG for each of the five projects and the monetary cost

Validity of the findings

Robust and comprehensive data

Additional comments

As travel (and plane) was the most impactful for GG emissions, could future calculations for conferences just focus on plane travel and offsets?

Suggest some discussion on mandatory compared to voluntary measurement of carbon footprint for events. This was voluntary, and I would be interested in how many hours (and estimated cost) to design and measure the impacts and implement the offset, ie, is it simple or complicated?

Some discussion on mandatory versus voluntary payment of offsets. See this paper for challenges with who pays for offsets https://www.mdpi.com/2071-1050/16/24/11019

Recommendations - should all future conferences include a carbon offset equivalent of 1-2 tonnes of GG as part of the conference fee.

Recommendations - please discuss how to leverage carbon-neutral conferences, which are 5 days a year, to the scientists' activities for the other 360 days a year. Should we aim for carbon-neutral journals (PeerJ?), universities, expeditions, and research institutes?. Should scientists report on their individual carbon footprint (mine is 1.8 planets) as well as their H-index

Line 342- How were the 5 carbon offset programs selected? Was this selected by the participants from the conference (vote) or the authors?

Was there an accreditation or independent review of the calculations and offset?

Reviewer 2 ·

Basic reporting

no comment

Experimental design

no comment

Validity of the findings

-The authors should offer clear recommendations regarding the reduction of the carbon footprint associated with conferences and symposia
- The authors should recommend nature-based solutions for carbon offsetting at the symposium.

Additional comments

The authors may consider future research aimed at transforming the ICRS into a net-zero emission conference.

---

## Round 0.2 · accepted · Accept

Thank you for incorporating the feedback from the reviewer(s) in this version of the manuscript. I have reviewed the new materials and reviews and find the manuscript to be ready for publication. Congratulations!

·

Basic reporting

-

Experimental design

-

Validity of the findings

-

Additional comments

The article has improved significantly and is suitable for publication.